# Unsupervised Emergence of Egocentric Spatial Structure from Sensorimotor Prediction

**Alban Laflaquière**
AI Lab, SoftBank Robotics Europe
Paris, France
alaflaquiere@softbankrobotics.com

**Michael Garcia Ortiz**
AI Lab, SoftBank Robotics Europe
Paris, France
mgarciaortiz@softbankrobotics.com

## Abstract

Despite its omnipresence in robotics application, the nature of spatial knowledge and the mechanisms that underlie its emergence in autonomous agents are still poorly understood. Recent theoretical works suggest that the Euclidean structure of space induces invariants in an agent's raw sensorimotor experience. We hypothesize that capturing these invariants is beneficial for sensorimotor prediction and that, under certain exploratory conditions, a motor representation capturing the structure of the external space should emerge as a byproduct of learning to predict future sensory experiences. We propose a simple sensorimotor predictive scheme, apply it to different agents and types of exploration, and evaluate the pertinence of these hypotheses. We show that a naive agent can capture the topology and metric regularity of its sensor's position in an egocentric spatial frame without any a priori knowledge, nor extraneous supervision.

## 1   Introduction

Current model-free Reinforcement Learning (RL) approaches have proven to be very successful at solving difficult problems, but seem to lack the ability to extrapolate and transfer already acquired knowledge to new circumstances [7, 33]. One way to overcome this limitation would be for learning agents to abstract from the data a model of the world that could support such extrapolation. For agents acting in the world, such an acquired model should include a concept of *space*, such that the spatial properties of the data they collect could be disentangled and extrapolated upon.

This problem naturally raises the question of the nature of space and how this abstract concept can be acquired. This question has already been addressed philosophically by great minds of the past [18, 36, 31], among which the approach proposed by H.Poincaré is of particular interest, as it naturally lends itself to a mathematical formulation and concrete experimentation. He was interested in understanding why we perceive ourselves as being immersed in a 3D and isotropic (Euclidean) space when our actual sensory experiences live in a multidimensional space of a different nature and structure (for instance, when the environment is projected on the flat heterogeneous surface of our retina). He suggested that the concept of space emerges via the discovery of *compensable sensory changes* that are generated by a change in the environment but can be canceled-out by a motor change. This *compensability* property applies specifically to *displacements* of objects in the environment and of the sensor, but not to non-spatial changes (object changing color, agent changing its camera aperture...). For instance, one can compensate the sensory change due to an object moving 1 meter away by moving 1 meter toward the object. Moreover, this compensability property is invariant to the content of the environment, as the displacement of an object can be compensated by the same motor change regardless of the type of the object. One can thus theoretically derive from the structure underlying these compensatory motor changes a notion of space abstracted from the specific sensory inputs that any given environment's content induces.

This philosophical stance has inspired recent theoretical works on the perception of space, and has in particular been coupled with the SensoriMotor Contingencies Theory (SMCT), a groundbreaking theory of perception that gives prominence to the role of motor information in the emergence of perceptive capabilities [32]. It led to theoretical results regarding the extraction of the dimension of space [23], the characterization of displacements as compensable sensory variations [38], the grounding of the concept of *point of view* in the motor space [24, 25], as well as the characterization of the metric structure of space via sensorimotor invariants [26]. These theoretical works suggest that an egocentric concept of space should emerge first, and that it could be grounded in the motor space as a way to economically capture the sensorimotor invariants that space induces. Our goal is thus to study how an unsupervised agent can build an internal representation of its sensor's egocentric spatial configuration akin to the $(x, y, z)$ Euclidean description that would otherwise be provided by a hand-designed model. This implies capturing the topology and regular metric structure of the external space in which the sensor moves in a way that does not depend on the content of the environment. This basic egocentric representation would be a solid foundation for the development of richer spatial knowledge and reasoning (ex: navigation, localization...).

The contribution of this work is to cast the aforementioned theoretical works in an unsupervised (self-supervised) Machine Learning frame. We further develop the formalization of space-induced sensorimotor invariants, and show that a representation capturing the topological and metric structure of space can emerge as a by-product of sensorimotor prediction. These results shed new light on the fundamental nature of spatial perception, and give some insight on the autonomous grounding of spatial knowledge in a naive agent's sensorimotor experience.

## 2 Related work

Only a quick overview of the large literature related to spatial representation learning is given here, leaving aside approaches where the spatial structure of the problem is largely hard-coded [6].

The problem is often conceptualized as the learning of grid or place cells, inspired by neuroscience [4]. Place cells have been built as a way to compress sensory information [2], or to improve sensorimotor and reward predictability [42, 37, 11]. Grid cells have been built as an intermediary representations in recurrent networks trained to predict an agent's position [3, 8]. Both place and grid cells have also been extracted by processing the internal state of a reservoir [1]. Representations akin to place cells and displacements have also been built from low-level sensorimotor interaction [21]. Theses works rely on the extraneous definition of "spatial" inductive bias or hand-designed loss functions.

In RL, state representation learning is often used to solve spatial tasks (ex: navigation). Some noteworthy works build states based on physical priors [16], system controllability [40], action sequencing [5], or disentanglement of controllable factors in the data [39]. Many end-to-end approaches are also applied to spatial problems without explicitly building spatial representations [29, 17], although auxiliary tasks are sometimes used to induce spatial constraints during training [28]. These works once again rely on hand-designed priors to obtain spatial-like representations.

Like in this work, forward sensorimotor predictive models are learned in many methods to compress sensory inputs, improve policy optimization, or derive a curiosity-like reward [12, 9, 41, 34]. Such forward models are also at the core of body schema learning approaches [15, 22]. Closer to this work, an explicit representation of displacements is built in [10] by integrating motor sequences for sensory prediction. However these works do not study how spatial structure can be captured in such models. Different flavors of Variational Auto-Encoders have been used to encode "spatial" factors of variation in a latent representation [13]; a work that interestingly led to the definition of disentanglement of spatial factors as *invariants* in an agent's experience [14]. These works however ignore the motor component of the problem.

Finally, this work is in line with the theoretical developments of [35, 23–25, 38, 26, 27], which address the fundamental problem of space perception in the framework of the SMCT, but frame them in an unsupervised machine learning framework. We show that the structure of space can get naturally captured as a by-product of sensorimotor prediction.

## 3 Problem setup

Let's consider an agent and an environment immersed in space. The agent has a fixed base, and is equipped with a sensor that it can move to explore its environment. It has only access to raw

sensorimotor experiences, and has no a priori knowledge about the world and the external space. Let $\mathbf{m} \in \mathbb{R}^{N_m}$ be the static configuration of its motors[1], referred to as *motor state*. Let $\mathbf{s} \in \mathbb{R}^{N_s}$ be the reading of its exteroceptive sensor, referred to as *sensory state*. Let $\epsilon \in \mathbb{R}^{N_\epsilon}$ be the state of the environment defining both its spatial and non-spatial properties. Finally, let $\mathbf{p} \in \mathbb{R}^{N_p}$ be the external position of the sensor in an egocentric frame of reference centered on the agent's base. This space of positions is assumed to be a typical Euclidean space with a regular topology and metric. Our goal is to build, from raw sensorimotor experiences $(\mathbf{m}, \mathbf{s})$, an internal representation $\mathbf{h} \in \mathbb{R}^{N_h}$ which captures the topological and metric structure of $\mathbf{p} \in \mathbb{R}^{N_p}$.

Inspired by Poincaré's original insight and borrowing from the formalism of [35], we assume that the agent's sensorimotor experience can be modeled as a continuous mapping parametrized by the state of the environment: $\mathbf{s} = \phi_\epsilon(\mathbf{m})$. The mapping $\phi$ represents all the constraints that the unknown structure of the world imposes on the agent's experience. In particular, it incorporates the structure of the space in which the agent and the environment are immersed. It has been shown that this structure manifests itself as invariants in the sensorimotor experience [26]. We reformulate here these invariants in a more compact way, taking advantage of the continuity of the sensorimotor mapping $\phi$. An intuitive description of them is given below, with a more complete mathematical derivation in Appendix A.

**Topological invariants:** The topology (and in particular the dimensionality) of $\mathbb{R}^{N_p}$ is a priori different from the one of $\mathbb{R}^{N_m}$ and $\mathbb{R}^{N_s}$. Yet, assuming no consistent sensory ambiguity between different sensor positions in the environments the agent explores, the sensory space experienced by the agent in each environment is a manifold, embedded in $\mathbb{R}^{N_s}$, which is homeomorphic to the space $\mathbb{R}^{N_p}$. Intuitively, this means that small displacements of the sensor are associated with small sensory changes, and vice versa, for any environmental state $\epsilon$. From a motor perspective, this implies that motor changes associated with small sensory changes correspond to small external displacements:

$$\forall \epsilon, \ |\phi_\epsilon(\mathbf{m}_t) - \phi_\epsilon(\mathbf{m}_{t+1})| \ll \mu \Leftrightarrow |\mathbf{s}_t - \mathbf{s}_{t+1}| \ll \mu \Leftrightarrow |\mathbf{p}_t - \mathbf{p}_{t+1}| \ll \mu, \quad (1)$$

where $|.|$ denotes a norm, and $\mu$ is a small value. The topology of $\mathbb{R}^{N_p}$ is thus accessible via sensorimotor experiences, and constrains how different motor states get mapped to similar sensory states. In the particular case of a redundant motor system, the multiple $\mathbf{m}$ which lead to the same sensor position $\mathbf{p}$ all generate the same sensory state $\mathbf{s}$ for any environmental state $\epsilon$. The agent has thus access to the fact that the manifold of sensory states, and thus the space of sensor positions, is of lower dimension than the one of its motor space. Note that these relations are invariant to the environmental state $\epsilon$.

We hypothesize that these *topological invariants* should be accessible to the agent under **condition I**: when exploring the world, the agent should experience *consistent* sensorimotor transitions $(\mathbf{m}_t, \mathbf{s}_t) \rightarrow (\mathbf{m}_{t+1}, \mathbf{s}_{t+1})$ such that the state of the environment $\epsilon$ stays unchanged during the transition.

**Metric invariants:** The metric of $\mathbb{R}^{N_p}$ is a priori different from the metric of $\mathbb{R}^{N_m}$ and $\mathbb{R}^{N_s}$. Yet, the metric regularity of the external space is accessible in the sensorimotor experience if the environment also undergoes displacements [26]. Indeed, let's consider two different sensory states $\mathbf{s}_t$ and $\mathbf{s}_{t+1}$ associated with two motor states $\mathbf{m}_t$ and $\mathbf{m}_{t+1}$ when the environment is in a first position $\epsilon$. The same sensory states can be re-experienced with two different motor states $\mathbf{m}_{t'}$ and $\mathbf{m}_{t'+1}$ after the environment moved rigidly to a new position $\epsilon'$. This is the *compensability* property of displacements coined by H.Poincaré. Thus, the consequence of the environment moving rigidly relatively to the agent's base is that equivalent displacements of the sensor in the external Euclidean space $\overrightarrow{\mathbf{p}_t\mathbf{p}_{t+1}} = \overrightarrow{\mathbf{p}_{t'}\mathbf{p}_{t'+1}}$ can generate the same sensory change $\mathbf{s}_t \rightarrow \mathbf{s}_{t+1}$ for different positions of the environment. In turn, the different motor changes $\mathbf{m}_t \rightarrow \mathbf{m}_{t+1}$ and $\mathbf{m}_{t'} \rightarrow \mathbf{m}_{t'+1}$ generating equivalent sensor displacements are associated with the same sensory changes for different positions of the environment. It ensues that (see Appendix A for the complete development):

$$\forall \epsilon, \epsilon', \begin{cases} |\phi_\epsilon(\mathbf{m}_t) - \phi_{\epsilon'}(\mathbf{m}_{t'})| \ll \mu \\ |\phi_\epsilon(\mathbf{m}_{t+1}) - \phi_{\epsilon'}(\mathbf{m}_{t'+1})| \ll \mu \end{cases} \Leftrightarrow |(\mathbf{p}_{t+1} - \mathbf{p}_t) - (\mathbf{p}_{t'+1} - \mathbf{p}_{t'})| \ll \mu. \quad (2)$$

These relations are once again invariant to the states $\epsilon$ and $\epsilon'$, as long as $\epsilon \rightarrow \epsilon'$ corresponds to a global rigid displacement of the environment. The metric regularity of $\mathbb{R}^{N_p}$ is thus accessible via sensorimotor experiences, and constrains how different motor changes get mapped to similar sensory changes, for different positions of the environment.

We hypothesize that these *metric invariants* should be accessible to the agent under **condition II**: the agent should experience *displacements of the environment $\epsilon \to \epsilon'$ in-between* consistent sensorimotor transitions $(\mathbf{m}_t, \mathbf{s}_t) \to (\mathbf{m}_{t+1}, \mathbf{s}_{t+1})$ and $(\mathbf{m}_{t'}, \mathbf{s}_{t'}) \to (\mathbf{m}_{t'+1}, \mathbf{s}_{t'+1})$.

Space thus induces an underlying structure in the way motor states map to sensory states. Interestingly, the sensory component of this structure varies for different environments, but not the motor one: a single $\mathbf{m}$ is always associated with the same egocentric position $\mathbf{p}$, regardless of the environmental state $\epsilon$, while the associated $\mathbf{s}$ changes with $\epsilon$. A stable representation of $\mathbf{p}$ can then be grounded in the motor space (as already argued in [25]), and shaped by sensory experiences which reveal this structure.

In this work, we hypothesize that capturing space-induced invariants is beneficial for sensorimotor prediction, as they can act as acquired priors over sensorimotor transitions no yet experienced. For instance, imagine two motor states $\mathbf{m}_a$ and $\mathbf{m}_b$ have always been associated with identical sensory states in the past. Then encoding them with the same representation $\mathbf{h}_a = \mathbf{h}_b$ can later help the agent extrapolate that if $\mathbf{m}_a$ is associated with a previously unseen sensory state $\mathbf{s}_a$, then $\mathbf{m}_b$ will also be. Therefore, we propose to train a neural network to perform sensorimotor prediction, and to analyze how it learns to encode motor states depending on the type of exploration that generates the sensorimotor data. We expect this learned representation to capture the topology of $\mathbf{p} \in \mathbb{R}^{N_p}$ when condition I is fulfilled, and to capture its metric regularity when condition II is fulfilled, without extraneous constraint nor supervision.

## 4  Experiments

**Sensorimotor predictive network:**   We propose a simple neural network architecture to perform sensorimotor prediction. The network's objective is to predict the sensory outcome $\mathbf{s}_{t+1}$ of a future motor state $\mathbf{m}_{t+1}$, given a current motor state $\mathbf{m}_t$ and sensory state $\mathbf{s}_t$. Additionally, we want both motor states $\mathbf{m}_t$ and $\mathbf{m}_{t+1}$ to be encoded in the same representational space. As illustrated in Fig. 1, the network is thus made of two types of modules: i) Net$_{\text{enc}}$, a Multi-Layer Perceptron (MLP) taking a motor state $\mathbf{m}_t$ as input, and outputting a motor representation $\mathbf{h}_t$ of dimension $N_h$, and ii) Net$_{\text{pred}}$, a MLP taking as input the concatenation of a current representation $\mathbf{h}_t$, a future representation $\mathbf{h}_{t+1}$, and a current sensory state $\mathbf{s}_t$, and outputting a prediction for the future sensory state $\tilde{\mathbf{s}}_{t+1}$. The overall network connects a predictive module Net$_{\text{pred}}$ to two siamese copies of a Net$_{\text{enc}}$ module, ensuring that both $\mathbf{m}_t$ and $\mathbf{m}_{t+1}$ are encoded the same way. The loss function to minimize is the Mean Squared Error (MSE) between the sensory prediction and the ground truth:

$$\text{Loss} = \frac{1}{K} \sum_{k=1}^{K} |\tilde{\mathbf{s}}_{t+1}^{(k)} - \mathbf{s}_{t+1}^{(k)}|^2, \tag{3}$$

where $K$ is the number of sensorimotor transitions collected by the agent. No extra component is added to the loss regarding the structure of the representation $\mathbf{h}$. Unless stated otherwise, the dimension $N_h$ is arbitrarily set to 3 for the sake of visualization. A more thorough description of the network and training procedure is available in Appendix B.

**Analysis of the motor representation:**   We use two measures $D_{topo}$ and $D_{metric}$ to assess how much the structure of the representation $\mathbf{h}$ built by the network differs from the one of the sensor position $\mathbf{p}$. The first corresponds to an estimation of the topological dissimilarity between a set in $\mathbb{R}^{N_p}$ and the corresponding set in $\mathbb{R}^{N_h}$:

$$D_{topo} = \frac{1}{N^2} \sum_{i,j}^{N} \frac{\left|\mathbf{h}_i - \mathbf{h}_j\right|}{\max_{kl}\left(\left|\mathbf{h}_k - \mathbf{h}_l\right|\right)} . \exp\left(\frac{-\alpha.\left|\mathbf{p}_i - \mathbf{p}_j\right|}{\max_{kl}\left(\left|\mathbf{p}_k - \mathbf{p}_l\right|\right)}\right), \tag{4}$$

where $|.|$ denotes the Euclidean norm, $N$ is the number of samples in each set, and $\alpha$ is arbitrarily set to 50. This measure is large when close sensor positions $\mathbf{p}$ are encoded by distant motor representations $\mathbf{h}$, and small otherwise.

The second measure corresponds to an estimation of the metric dissimilarity between those same two sets. At this point, it is important to notice that the metric invariants described in Sec. 3 only imply relative distance constraints (see also [26]). Consequently, any representation $\mathbf{h}$ related to $\mathbf{p}$ via an affine transformation would respect these constraints[2]. In order to properly assess if the two sets share

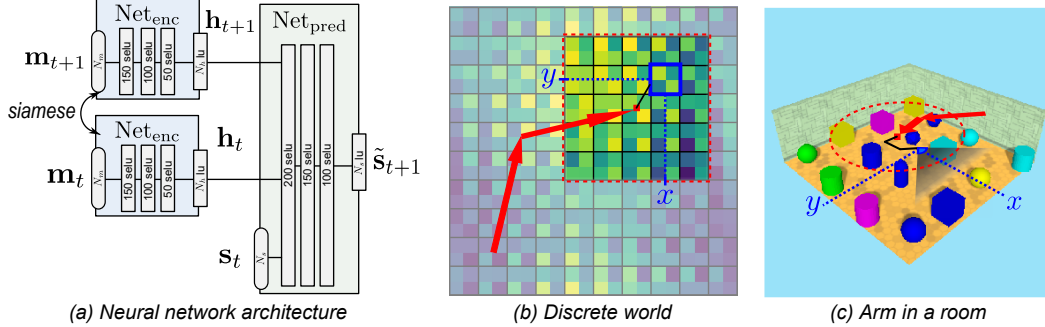

| (a) Neural network architecture | (b) Discrete world | (c) Arm in a room |

Figure 1: (a) The neural network architecture featuring two siamese instances of the $\text{Net}_{\text{enc}}$ module, and a $\text{Net}_{\text{pred}}$ module. (b-c) Illustrations of the Discrete world and Arm in a room simulations. The fixed base of the agent is represented by a red square. The working space reachable by the sensor is framed in dotted red. The sensor and its current egocentric position $\mathbf{p} = [x, y]$ in this frame are displayed in blue. The environment can translate relatively to the agent's base, which is equivalent to an (opposite) displacement of the agent's base itself, as illustrated by red arrows. (Best seen in color)

the same metric regularity, we first perform a linear regression of $\mathbf{p}$ on $\mathbf{h}$ to cancel out the potential affine transformation between the two. We denote $\mathbf{h}^{(p)} = A.\mathbf{h} + b$ the resulting projection of $\mathbf{h}$ in $\mathbb{R}^{N_p}$, where $A$ and $b$ are the optimal parameters of the linear regression. The second dissimilarity $D_{metric}$ is then defined as:

$$D_{metric} = \frac{1}{N^2} \sum_{i,j}^{N} \frac{\left| \left|\mathbf{h}_i^{(p)} - \mathbf{h}_j^{(p)}\right| - \left|\mathbf{p}_i - \mathbf{p}_j\right| \right|}{\max_{kl}\left(\left|\mathbf{p}_k - \mathbf{p}_l\right|\right)}. \tag{5}$$

This measure is large when the distance between two sensor positions $\mathbf{p}$ differs from the distance between the two corresponding motor representations $\mathbf{h}$ (after affine projection), and is small otherwise. It is equal to zero when there exists a perfect affine mapping between $\mathbf{h}$ and $\mathbf{p}$; in which case the two sets have equivalent metric regularities.

Note that in (4) and (5), distances are normalized by the largest distance in the corresponding space in order to avoid undesired scaling effects. In the following, the dissimilarities are computed on sets of $\mathbf{p}$ and corresponding $\mathbf{h}$ generated by sampling the motor space in a fixed and regular fashion (see Fig. 3). This ensures a rigorous comparison of their values between epochs and between runs.

**Types of exploration:**   Three types of exploration of the environment are considered to test the hypotheses laid out in Sec. 3. They correspond to different ways to generate the sensorimotor transitions $(\mathbf{m}_t, \mathbf{s}_t) \rightarrow (\mathbf{m}_{t+1}, \mathbf{s}_{t+1})$ fed to the network during training:

*Inconsistent transitions in a moving environment*: The motor space is randomly sampled, and the environment randomly moves between $t$ and $t + 1$. Both conditions I and II are broken, as the agent explores a constantly moving environment, such that its sensorimotor transitions have no spatiotemporal consistency[3]. We refer to this type of exploration as **MEM** (Motor-Environment-Motor), in agreement with the order of changes for each transition.

*Consistent transitions in a static environment*: The motor space is sampled randomly, and the environment stays static. Condition I is fulfilled, as the agent experiences spatiotemporally consistent transitions in a static environment, but not condition II, as the environment does not move between transitions. We refer to this type of exploration as **MM** (Motor-Motor).

*Consistent transitions in a moving environment*: The motor space is randomly sampled, and the environment randomly moves after each transition (after $t + 1$). Both conditions I and II are fulfilled, as the agent experiences spatiotemporally consistent transitions, and the environment moves between transitions. We refer to this type of exploration as **MME** (Motor-Motor-Environment).

Additional details on the sampling procedure are available in Appendix B. According to Sec. 3, we expect the sensorimotor data to contain no spatial invariants in the MEM case, topological invariants in the MM case, and both topological and metric invariants in the MME case.

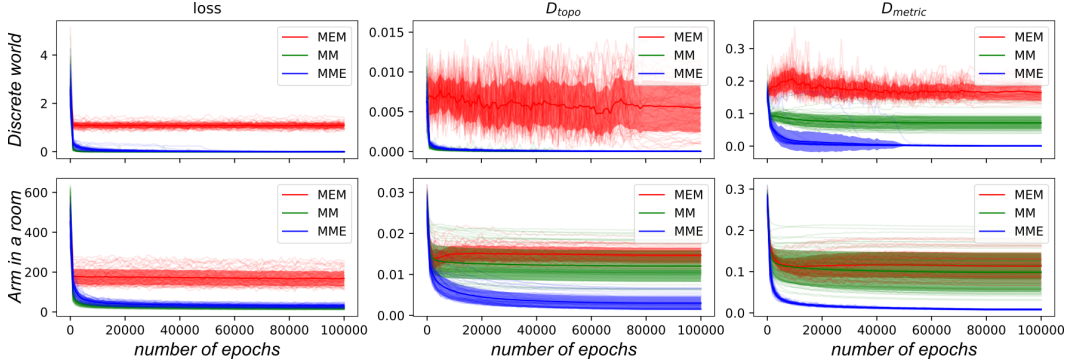

Figure 2: Evolution of the loss and the dissimilarity measures $D_{topo}$ and $D_{metric}$ during training for both setups, and for the three types of exploration. The displayed means and standard deviations are computed over 50 independent runs. (Best seen in color)

**Agent-Environment setups:** We simulate two different agent-environment setups: *Discrete world* and *Arm in a room*. The first one is a minimalist artificial setup designed to test our hypotheses in optimal conditions. It corresponds to a 2D grid-world environment that the agent can explore by generating 3D motor states $\mathbf{m}$ which map, in a non-linear and redundant way, to 2D positions $\mathbf{p}$ of the sensor in the grid (see Fig. 1(b)). For each position the agent receives a 4D sensory input $\mathbf{s}$ that is designed to vary smoothly over the grid and to have no sensory ambiguity between different positions. The whole grid can translate with respect to the agent's base, changing its state $\epsilon$, and acts as a torus to avoid any border effect.

The second one is a more complex and realistic setup in which a three-segment arm equipped with a camera explores a 3D room filled with random objects (see Fig. 1(c)). The agent can change its camera position $\mathbf{p}$ in a horizontal plane (with a fixed orientation) by generating 3D motor states $\mathbf{m}$, and receives sensory inputs $\mathbf{s}$ of size 768 ($16 \times 16$ RGB pixels). The whole room can translate in 2D with respect to the agent's base, changing its state $\epsilon$, and the arm cannot move outside of the room. A more complete description of the simulations is available in Appendix C.

## 5 Results

We evaluate the three types of exploration on the two experimental setups. Each simulation is run 50 times, with all random parameters drawn independently on each trial. During training, the measures $D_{topo}$ and $D_{metric}$ are evaluated on a fixed regular sampling of the motor space. Their evolution, as well as the evolution of the loss, are displayed in Fig. 2. Additionally, Fig. 3 shows the final representations $\mathbf{h}$ of the regular motor sampling, for one randomly selected trial of each simulation. The corresponding positions $\mathbf{p}$ and the projection $\mathbf{h}^{(p)}$ are displayed in the same space in order to visualize how much their metric structures differ.

**Discrete world results:** The results clearly show an impact of the type of exploration on the motor encoding built by the network. First of all, as expected, the loss is high in the MEM case because the constant movements of the environment prevent any accurate sensorimotor prediction. On the contrary, it is low in the MM and MME cases, as the consistency of transitions enables accurate sensorimotor prediction (upper-bounded by the expressivity power of the Net$_{pred}$ module).

More interestingly, the topological dissimilarity $D_{topo}$ has a significantly smaller value and variance in the MM and MME cases than in the MEM case. This seems to indicate that the MEM exploration leads to arbitrary representations, while the topologies of $\mathbf{h}$ and $\mathbf{p}$ are more similar when the agent can experience consistent sensorimotor transitions $(\mathbf{m}_t, \mathbf{s}_t) \rightarrow (\mathbf{m}_{t+1}, \mathbf{s}_{t+1})$ during which the environment does not move.

Similarly, the metric dissimilarity $D_{metric}$ is high in the MEM case, average in the MM case, and low in the MME case. This intermediate value in the MM case is due to the fact that capturing the topology of $\mathbf{p}$ also indirectly reduces the metric dissimilarity. However, the very low value in the MME case seems to indicate that the metric of $\mathbf{h}$ displays a regularity which is similar to the one of $\mathbf{p}$. This regularity is thus captured only when the agent can experience movements of the environment

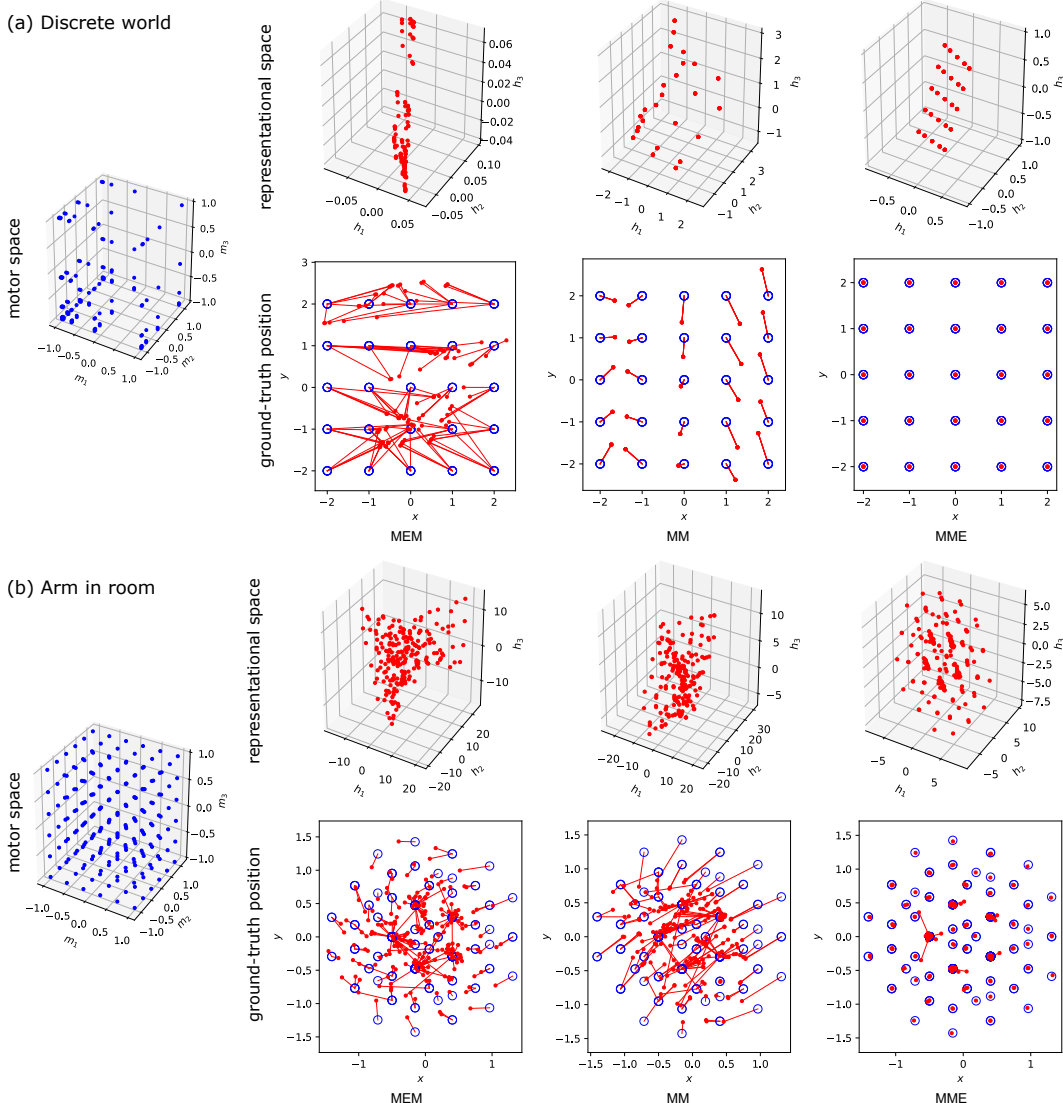

Figure 3: Visualization of the normalized regular motor sampling $\mathbf{m}$ (blue dots), its representation $\mathbf{h}$ in the representational space (red dots), and the corresponding ground truth position $\mathbf{p}$ (blue circles) for the three types of exploration, and for both the Discrete world (a) and Arm in a room (b) setups. The linear regression $\mathbf{h}^{(p)}$ of the representations are also displayed in the space of positions. Lines have been added to visualize the distances between each $\mathbf{h}^{(p)}$ and its ground truth counterpart $\mathbf{p}$. In (a), the regular (but non-linear) 3D motor sampling generates a regular 2D grid of positions, where each position corresponds to 5 redundant motor states. In (b), the regular 3D motor sampling generates a star-shaped set of positions, where some positions, like the inner corners of the star, correspond to multiple motor states. (Best seen in color)

between consistent sensorimotor transitions.

This analysis is confirmed in Fig. 3 where $\mathbf{h}$ and its affine projection $\mathbf{h}^{(p)}$ in $\mathbb{R}^{N_p}$ display an arbitrary structure in the MEM case, a structure topologically equivalent to the one of $\mathbf{p}$ in the MM case, and a structure topologically and metrically equivalent to the one of $\mathbf{p}$ in the MME case.

**Arm in a room results:** The results in this more complex and realistic setup are globally similar to the previous ones, which shows a consistency in the phenomenons we just described. However, two important differences can be pointed out.

The first is that, in the MEM case, $D_{topo}$ and $D_{metric}$ seem to decrease more than what would

be expected for an arbitrarily random representation $\mathbf{h}$. We argue that this phenomenon is due to a "border effect" induced by the walls of the room. Indeed, as the sensor's and environment's displacements are limited, each motor state $\mathbf{m}$ is statistically associated with a different distribution of sensory states over the whole course of the exploration. For instance, a motor state corresponding to the arm extended to the right will never experience the sensory states corresponding to the sensor being at the far-left of the room, and inversely. As the sensory distribution associated with $\mathbf{m}$ varies smoothly with $\mathbf{p}$, the agent can indirectly infer the topology of $\mathbb{R}^{N_p}$ from it. We can indeed see in Fig. 3 that $\mathbf{h}$ (or even better $\mathbf{h}^p$) tends to capture the topology of $\mathbf{p}$ in the MEM case, although with less accuracy than in the MM and MEM cases. Note that this was not the case in the previous torus-like grid world as any motor state could be associated with any square of the grid over the course of the exploration.

The second difference is that $D_{topo}$ is higher in the MM case than in the previous setup. After an empirical visualization of the environments and associated learned representations $\mathbf{h}$, we argue that this is due to potential sensory ambiguity. Indeed, the random room the agent explores can present ambiguities, such that very different positions of the sensor can be associated with very similar sensory inputs. When this happens, the representation $\mathbf{h}$ built by the network can arbitrarily encode the same way different motor states associated with these different positions. This leads to a representation manifold that is non-trivially twisted in $\mathbb{R}^{N_h}$ (see Fig. 3(b)), and degrades the measure $D_{topo}$. Note that this sensory ambiguity is not an issue in the MME case anymore, as the movements of the environment help disambiguate these different motor states.

The same experiments have been run with a representational space of dimension $N_h = 25$ and with more complex agent morphologies, and led to qualitatively similar results. This seems to indicate that the capture of the topological and metric invariants is insensitive to the dimension of $\mathbf{h}$ and the complexity of the forward mapping. A more detailed analysis of the all these different simulations results is available in Appendix D and E.

# 6    Conclusion

We addressed the problem of the unsupervised grounding of the concept of space in a naive agent's sensorimotor experience. Inspired by previous philosophical and theoretical work, we argue that such a notion should first emerge as a basic representation of the egocentric position of a sensor moving in space. We showed that the structure of the Euclidean space, in which the agent is immersed alongside its environment, induces sensorimotor invariants. They constrain the way motor states gets mapped to sensory inputs, independently from the actual content of the environment, and carry information about the topology and metric of the external space. This structure can potentially be extracted from the sensorimotor flow to build an internal representation of the sensor egocentric position, grounded in the motor space, and abstracted from the content of the environment and the specific sensory states it induces. We hypothesized that capturing space-induced invariants is beneficial for sensorimotor prediction. As a consequence, topological and metric invariants should naturally be captured by a network learning to perform sensorimotor prediction. We proposed such a network architecture, and designed different types of exploration of the environment such that topological and metric invariants were present or not in the resulting sensorimotor transitions fed to the network. We tested two different simulated agent-environment setups, and showed that when spatial invariants are present in the sensorimotor data, they get naturally captured in the internal motor representation built by the agent. So, when the agent can experience consistent sensorimotor transitions during which the environment does not change, the internal motor representation captures the topology of the external space in which its sensor is moving. Even more interestingly, when the agent can also experience displacements of the environment between its consistent sensorimotor transitions, the internal motor representation captures the metric regularity of this external space. These results thus suggest that the concept of an external Euclidean space, although still in its most basic form here, could emerge in a situated agent as a by-product of learning to predict its sensorimotor experience.

We hope this work can be a stepping stone for further extensions of the approach and the unsupervised acquisition of richer spatial knowledge. A first obvious step will be to extend the exploration to 3 translations and 3 rotations of the sensor in space. A second very important step will be to derive, from the current basic egocentric spatial representation, an allocentric representation in which the spatial configurations of external objects could also be characterized; a problem for which H.Poincaré also had some interesting intuitions.

## Footnotes

[1] If the body is not controlled in position, $\mathbf{m}$ can be a proprioceptive reading of the body configuration.

[2]An affine transformation preserves topology and distance ratios.

[3]It is akin to the kind of data a passive and non-situated agent receives in typical machine learning settings.

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
