[Supplementary Material]

# Unsupervised Emergence of Egocentric Spatial Structure from Sensorimotor Prediction - Appendix

**Alban Laflaquière**
AI Lab, SoftBank Robotics Europe
Paris, France
alaflaquiere@softbankrobotics.com

**Michael Garcia Ortiz**
AI Lab, SoftBank Robotics Europe
Paris, France
mgarciaortiz@softbankrobotics.com

## A Mathematical formalism

### A.1 Sensorimotor interaction

We consider the sensorimotor interaction between an agent and its environment immersed in an Euclidean space. The agent has a base with a fixed position in space, and is equipped with an exteroceptive sensor that it can move in space thanks to its motors. These motors are supposed to be controlled in position, which means that each motor state corresponds to a fixed posture of the agent, and thus to a fixed position of the sensor in space. The environment itself can move in a rigid fashion, independently from the agent (see Fig. 4).

In this work, we show how performing sensorimotor prediction naturally leads such an agent to build an internal representation of the egocentric position of its sensor which captures both the topology and metric regularity of the external space.

Figure 4: Diagram of a three-segment agent (black) with a fixed base (red square) exploring an environment (brown) that can move. The motor state $\mathbf{m}$ defines the configuration of the arm. It is associated with a position $\mathbf{p} = [x, y]$ of its end-effector sensor (blue), relative to its base. This egocentric frame of reference is represented as a grid centered on the agent's basis. The sensor generates a sensory state $\mathbf{s}$. The agent can move its sensor in the environment, but the environment can also move (red arrow) relatively to the agent's basis, changing its state $\epsilon$. Note that such a rigid displacement of the whole environment is equivalent to an opposite displacement of the agent's basis in a static environment (red dotted arrow). (Best seen in color)

Figure 5: Illustration of the variables and mappings involved in the formalism. (Best seen in color)

Note that, from a sensorimotor perspective, a displacement of the environment relative to the (fixed) agent's base is equivalent to a movement of the agent's base relative to the (fixed) environment. This equivalence is behind the concept of *compensability* introduced by Poincaré to characterize spatial interactions [36]. Considering that the agent moves its base relative to a static environment sounds like a more natural description, as we rarely experience rigid displacements of the whole environment around us. However we favor the first description in this work, as a way to follow Poincaré's original insight and to better emphasize the interaction between motor changes and environmental changes in the mathematical formalism.

## A.2 Variables and mappings

We denote $\mathbf{m}$ the agent's motor state, which corresponds to a static posture of its body, and $\mathcal{M}$ the set of all $\mathbf{m}$. We denote $\mathbf{p}$ the egocentric position of the sensor relative to the agent's base, and $\mathcal{P}$ the set of all $\mathbf{p}$. We denote $\epsilon$ the state of the environment, which describes both its spatial and non-spatial properties, and $\mathcal{E}$ the set of all $\epsilon$. Finally, we denote $\mathbf{s}$ the agent's sensory state, which corresponds to an instantaneous reading of the sensor's output without transient phase, and $\mathcal{S}$ the set of all $\mathbf{s}$.

The different relations between these variables are illustrated in Fig.5. Each motor state $\mathbf{m}$ is associated with a sensor position $\mathbf{p}$ via a forward mapping denoted $f$:

$$\begin{aligned} f: \ & \mathcal{M} \to \mathcal{P} \\ & \mathbf{m} \mapsto f(\mathbf{m}) = \mathbf{p}. \end{aligned} \tag{1}$$

Similarly, each pair $(\mathbf{p}, \epsilon)$ of sensor position and environmental state is associated with a sensory state $\mathbf{s}$ via a sensory mapping denoted $g$:

$$\begin{aligned} g: \ & \mathcal{P} \times \mathcal{E} \to \mathcal{S} \\ & \mathbf{p}, \epsilon \mapsto g(\mathbf{p}, \epsilon) = \mathbf{s}. \end{aligned} \tag{2}$$

Due to the inherent properties of space, the sensory mapping $g$ has the following invariance property:

$$\forall \delta, \mathbf{p}, \epsilon, \ g(\mathbf{p}, \epsilon + \delta) = g(\mathbf{p} - \delta, \epsilon), \tag{3}$$

where $\delta$ represents a rigid displacement of either the whole environment or of the sensor. In other words, from a sensory perspective, a displacement of the environment is equivalent to an opposite displacement of the sensor.

In this work, we consider that the agent is naive. It only has direct access to the motor states it produces and to the related sensory states it receives: $(\mathbf{m}, \mathbf{s})$. It does not have a priori information about $f$, $g$, or $\epsilon$. From a sensorimotor perspective, the overall composite mapping $g \circ f$ can thus be re-expressed as a sensorimotor mapping $\phi_\epsilon$:

$$\mathbf{s} = g\big(f(\mathbf{m}), \epsilon\big) = \phi_\epsilon(\mathbf{m}), \tag{4}$$

where $\epsilon$ is treated as a parameter of the unknown mapping $\phi$ to emphasize that the agent has no information about both $\phi$ and $\epsilon$ (see Fig. 5).

## A.3 Hypotheses

Taking inspiration from the differential geometry-based formalism introduced in [35], we assume that $\mathcal{M} \subset \mathbb{R}^{N_m}$, $\mathcal{S} \subset \mathbb{R}^{N_s}$, $\mathcal{P} \subset \mathbb{R}^{N_p}$, and $\mathcal{E} \subset \mathbb{R}^{N_\epsilon}$ are manifolds embedded in their respective finite vector spaces. The variables $\mathbf{m}$, $\mathbf{s}$, $\mathbf{p}$, $\epsilon$ can thus be expressed as real-valued vectors. Additionally, we posit that $\mathcal{M}$ and $\mathcal{P}$ are convex bounded Euclidean subspaces of their respective dimensions, to take into account that the agent has a limited body and thus a limited working space in which it can move its sensor.

Moreover, we make the strong assumption that the mappings $f$, $g$ (and consequently $\phi$) are continuous, which means that an infinitesimally small change in their input leads to an infinitesimally small change in their output. Without any loss of generality, we assume that $f$ is surjective, in order to accommodate for redundant motor systems for which multiple motor states $\mathbf{m}$ can be associated with the same sensor position $\mathbf{p}$. Finally, for convenience, let $g_\epsilon$ denote the sensory mapping associated with a fixed environmental state $\epsilon$:

$$\forall \epsilon, g_\epsilon : \mathcal{P} \to \mathcal{S}$$
$$\mathbf{p} \mapsto g_\epsilon(\mathbf{p}) = g(\mathbf{p}, \epsilon) = \mathbf{s}. \tag{5}$$

In any environment, we assume that there is no sensory ambiguity between two different sensor positions. Formally, the mapping $g_\epsilon$ is thus assumed to be bijective[1] for all $\epsilon$:

$$\forall \epsilon, \mathbf{s}, \exists! \mathbf{p} \text{ such that: } g_\epsilon(\mathbf{p}) = \mathbf{s}, \tag{6}$$

making it a homeomorphism between $\mathcal{P}$ and $\mathcal{S}$, as it is also continuous. This means that, for a given state $\epsilon$ of the environment, each position of the sensor $\mathbf{p}$ is associated with a unique sensory state $\mathbf{s}$, and vice versa. This hypothesis is expected to hold if the sensor is rich enough and if the environment does not present symmetries. This apparently strong constraint of the model turns out to be relatively weak in practice. Indeed, thanks to the statistical machine learning approach used in this work, this non-ambiguity assumption has to hold statistically over all environmental states. In other words, it is sufficient that two sensor positions lead to different sensory states for at least *some* environmental states $\epsilon$, but not necessarily all of them. In strongly unfavorable scenarios, one could also consider extending the model by integrating sensorimotor experiences in time to avoid ambiguities.

## A.4 Sensorimotor invariants:

The manifold of sensor positions $\mathcal{P}$ has a priori a different topology and metric than the motor manifold $\mathcal{M}$. For instance, in the case of a redundant motor system, a single sensor position $\mathbf{p} \in \mathcal{P}$ is associated with a subset of dimension greater than 0 in $\mathcal{M}$. As a consequence, the dimension of $\mathcal{P}$ is lower than the one of $\mathcal{M}$ and their topologies differ. Similarly, the same change (same direction and amplitude) of sensor position in the external space is a priori associated with different motor changes (different directions and amplitudes), depending on the starting sensor position (and vice versa). The metrics of $\mathcal{P}$ and $\mathcal{M}$ thus differ in a non-linear way.

Yet, it has been shown in [26] that the topology and metric of $\mathcal{P}$ induce invariants in the sensorimotor experiences of the agent. Here, we rigorously reformulate these invariants, and extend them to small neighborhood by taking advantage of the assumption of continuity of $\phi$.

Given the previously stated properties of the mappings, we can generally write, for any rigid displacement $\delta$, that:

$$\forall \epsilon, \epsilon' = \epsilon + \delta, \quad \mathbf{s}_i = \mathbf{s}_{i'}$$
$$(4) \Leftrightarrow \phi_\epsilon(\mathbf{m}_i) = \phi_{\epsilon'}(\mathbf{m}_{i'})$$
$$(4) \Leftrightarrow g\big(f(\mathbf{m}_i), \epsilon\big) = g\big(f(\mathbf{m}_{i'}), \epsilon'\big)$$
$$(1) \Leftrightarrow g(\mathbf{p}_i, \epsilon) = g(\mathbf{p}_{i'}, \epsilon + \delta) \tag{7}$$
$$(3) \Leftrightarrow g(\mathbf{p}_i, \epsilon) = g(\mathbf{p}_{i'} - \delta, \epsilon)$$
$$(5) \Leftrightarrow g_\epsilon(\mathbf{p}_i) = g_\epsilon(\mathbf{p}_{i'} - \delta)$$
$$(6) \Leftrightarrow \mathbf{p}_i = \mathbf{p}_{i'} - \delta.$$

If both sensorimotor experiences $(\mathbf{m_i}, \mathbf{s_i})$ and $(\mathbf{m_{i'}}, \mathbf{s_{i'}})$ are collected in the same environmental state $\epsilon$, then $\delta = 0$ and we have:

$$\forall \epsilon, \mathbf{s}_i = \mathbf{s}_{i'} \Leftrightarrow \phi_\epsilon(\mathbf{m}_i) = \phi_\epsilon(\mathbf{m}_{i'}) \Leftrightarrow \mathbf{p}_i = \mathbf{p}_{i'}. \tag{8}$$

Figure 6: Diagram illustrating topological and metric invariants with a three-segment arm agent. (a) Two motor configurations $\mathbf{m}_i$ and $\mathbf{m}_{i'}$ generating similar sensory states $\mathbf{s}_i$ and $\mathbf{s}_{i'}$ are associated with similar egocentric positions $\mathbf{p}_i$ and $\mathbf{p}_{i'}$ of the sensor. (b) A displacement $\epsilon \rightarrow \epsilon'$ (red arrow) of the environment relative to the agent's base (red square) can be *compensated* by the agent. As a result, two motor states $\mathbf{m}_i$ and $\mathbf{m}_{i'}$ are associated with similar sensory states $\mathbf{s}_i$ and $\mathbf{s}_{i'}$ before and after the displacement. The same logic applies to $\mathbf{m}_j$ and $\mathbf{m}_{j'}$, and their similar sensory states $\mathbf{s}_j$ and $\mathbf{s}_{j'}$. This means that the 4 corresponding egocentric sensor positions $\mathbf{p}_i$, $\mathbf{p}_{i'}$, $\mathbf{p}_{j'}$, and $\mathbf{p}_j$ form a parallelogram in space. It is represented on the right, relatively to the agent's base, with colored double-headed arrows indicating metric equivalences of displacements between its vertices. Note that a displacement of the environment is equivalent to a displacement of the base of the agent (red dotted arrow). (Best seen in color)

This relation is invariant to the environmental state $\epsilon$. Through the sensory experiences collected when exploring environments, the agent can thus discover that different motor states are associated with the same external sensor position. The necessary condition for such a discovery is that the agent can explore more than one sensorimotor pair $(\mathbf{m_i}, \mathbf{s_i})$ for any environmental state $\epsilon$.

If we now consider not two but four sensorimotor pairs such that two of them $\{(\mathbf{m_i}, \mathbf{s_i}), (\mathbf{m_j}, \mathbf{s_j})\}$ are collected in a first environmental state $\epsilon$, and the two others $\{(\mathbf{m_{i'}}, \mathbf{s_{i'}}), (\mathbf{m_{j'}}, \mathbf{s_{j'}})\}$ are collected in a second environmental state $\epsilon'$, we can write:

$$\forall \epsilon, \epsilon' = \epsilon + \delta, \begin{cases} \mathbf{s}_i = \mathbf{s}_{i'} \\ \mathbf{s}_j = \mathbf{s}_{j'} \end{cases}$$

$$(4) \Leftrightarrow \begin{cases} \phi_\epsilon(\mathbf{m}_i) = \phi_{\epsilon'}(\mathbf{m}_{i'}) \\ \phi_\epsilon(\mathbf{m}_j) = \phi_{\epsilon'}(\mathbf{m}_{j'}) \end{cases} \tag{9}$$

$$(7) \Leftrightarrow \begin{cases} \mathbf{p}_i = \mathbf{p}_{i'} - \delta \\ \mathbf{p}_j = \mathbf{p}_{j'} - \delta \end{cases}$$

$$\Leftrightarrow \quad \mathbf{p}_j - \mathbf{p}_i = \mathbf{p}_{j'} - \mathbf{p}_{i'}.$$

This relation is once again invariant to the environmental states $\epsilon$ and $\epsilon'$, as long as $\delta$ corresponds to a rigid displacement in space. Through the sensory experience collected when exploring environments that can move, the agent can thus discover that different motor changes are associated with equivalent external displacements of the sensor. To be precise, $\mathbf{m}_i \rightarrow \mathbf{m}_j$ and $\mathbf{m}_{i'} \rightarrow \mathbf{m}_{j'}$ are associated with the same external displacement $\overrightarrow{\mathbf{p}_i\mathbf{p}_j} = \overrightarrow{\mathbf{p}_{i'}\mathbf{p}_{j'}}$. The necessary condition for such a discovery is that the agent explores more than one sensorimotor pair $(\mathbf{m_i}, \mathbf{s_i})$ in more than one environmental state $\epsilon$.

Finally, let's re-express equations (8) and (9) as follows:

$$\forall \epsilon, \; |\phi_\epsilon(\mathbf{m}_i) - \phi_\epsilon(\mathbf{m}_{i'})| = 0 \Leftrightarrow |\mathbf{p}_i - \mathbf{p}_{i'}| = 0, \tag{10}$$

$$\forall \epsilon, \epsilon' = \epsilon + \delta, \begin{cases} |\phi_\epsilon(\mathbf{m}_i) - \phi_{\epsilon'}(\mathbf{m}_{i'})| = 0 \\ |\phi_\epsilon(\mathbf{m}_j) - \phi_{\epsilon'}(\mathbf{m}_{j'})| = 0 \end{cases} \tag{11}$$

$$\Leftrightarrow \quad |(\mathbf{p}_j - \mathbf{p}_i) - (\mathbf{p}_{j'} - \mathbf{p}_{i'})| = 0,$$

where $|.|$ denotes a norm, and let's take advantage of the continuity of the mappings $f$, $g$, and $\phi$, to generalize them to local neighborhoods:

$$\forall \epsilon, \; |\phi_\epsilon(\mathbf{m}_i) - \phi_\epsilon(\mathbf{m}_{i'})| \ll \mu \Leftrightarrow |\mathbf{p}_i - \mathbf{p}_{i'}| \ll \mu, \tag{12}$$

Figure 7: Illustration of the variables processed by the neural architecture. (Best seen in color)

$$\forall \epsilon, \epsilon' = \epsilon + \delta, \begin{cases} |\phi_\epsilon(\mathbf{m}_i) - \phi_{\epsilon'}(\mathbf{m}_{i'})| \ll \mu \\ |\phi_\epsilon(\mathbf{m}_j) - \phi_{\epsilon'}(\mathbf{m}_{j'})| \ll \mu \end{cases} \tag{13}$$
$$\Leftrightarrow \quad |(\mathbf{p}_j - \mathbf{p}_i) - (\mathbf{p}_{j'} - \mathbf{p}_{i'})| \ll \mu,$$

where $\mu$ is a small value.

Due to their nature, we refer to the invariants of Equation (12) as *topological invariants*, and the ones of Equation (13) as *metric invariants*. A simple illustration of both types of invariants is proposed in Fig. 6.

## B  Neural network architecture and training

### B.1  Neural network

We propose a simple neural network architecture to perform sensorimotor prediction. It is composed of two types of module: $\text{Net}_{\text{enc}}$ and $\text{Net}_{\text{pres}}$.

The $\text{Net}_{\text{enc}}$ module projects a motor state $\mathbf{m}_t$ onto a representation $\mathbf{h}_t$ of dimension $N_h$. It consists of a fully connected Multi-Layer Perceptron (MLP) with three hidden layers of respective sizes $(150, 100, 50)$ with SeLu activation functions [20], and a final output layer of size $N_h$ with linear activation functions. This last layer corresponds to the space $\mathbb{R}^{N_h}$ in which the motor representation are analyzed in this work.

The $\text{Net}_{\text{pred}}$ module takes as input the concatenation $(\mathbf{h}_t, \mathbf{h}_{t+1}, \mathbf{s}_t)$ of a current motor representation, a future motor representation, and a current sensory state, and outputs a prediction $\tilde{\mathbf{s}}_{t+1}$ of the future sensory state $\mathbf{s}_{t+1}$. It consists of a fully connected MLP with three hidden layers of respective sizes $(200, 150, 100)$ with SeLu activation functions, and a final output layer of size $N_s$ with linear activation functions.

The overall network architecture connects the predictive module $\text{Net}_{\text{pred}}$ to two siamese copies of the $\text{Net}_{\text{enc}}$ module, ensuring that both motor states $\mathbf{m}_t$ and $\mathbf{m}_{t+1}$ are consistently encoded using the same mapping.

Note that the simulations have also be run with ReLu units [30] in place of the SeLu units and produced qualitatively identical results.

As illustrated in Fig. 7, a parallel can be drawn between the network illustrated in Fig.1 and the sensorimotor mapping illustrated in Fig. 5. Indeed, one can relate $\text{Net}_{\text{enc}}$ to the forward mapping $f$, and $\text{Net}_{\text{pred}}$ to the sensory mapping $g$. However, as $\epsilon$ is not directly accessible to the agent, its counterpart in the network corresponds to the current sensorimotor pair, where $\mathbf{m}_t$ is encoded as $\mathbf{h}_t$ (red frame in Fig. 7). As we assume the sensorimotor experience to be unambiguous, the sensorimotor experience $(\mathbf{m}_t, \mathbf{s}_t)$ indeed carries information about the state of the environment $\epsilon$.

Note that in agreement with the hypothesis of Sec. A and the fact that $\text{Net}_{\text{enc}}$ instantiates a continuous mapping, the set $\mathcal{H} \subset \mathbb{R}^{N_h}$ of all representations $\mathbf{h}$ is also assumed to be a manifold.

## B.2 Loss minimization

The unsupervised (or self-supervised) objective of the network is to minimize the MSE loss:

$$\text{Loss} = \frac{1}{K} \sum_{k=1}^{K} |\tilde{\mathbf{s}}_{t+1}^{(k)} - \mathbf{s}_{t+1}^{(k)}|^2,$$

where $k$ denotes a sample index, $K$ is the number of samples in the training dataset, and $|.|$ denotes the Euclidean norm. Importantly, no particular component is added to the loss function regarding the structure of the representation $\mathbf{h}$ built by the network. The loss is minimized using the ADAM optimizer [19], with a learning rate linearly decreasing from $10^{-3}$ to $10^{-5}$ in $8 \times 10^4$ epochs, and a mini-batch size of 100 sensorimotor transitions. The optimization is stopped after $10^5$ epochs (a single mini-batch is fed to the network at each epoch).

## B.3 Training data

The training data are generated by having the simulated agent explore its environment and collect sensorimotor transitions $(\mathbf{m}_t, \mathbf{s}_t) \rightarrow (\mathbf{m}_{t+1}, \mathbf{s}_{t+1})$. A total of 150000 transitions are collected for each simulation by randomly sampling pairs $(\mathbf{m}_t, \mathbf{m}_{t+1})$ in the motor space (uniform distribution) and collecting the corresponding sensory inputs. Depending on the type of exploration, the environment also translates during the data collection, instantaneously changing its position in the horizontal plane relative to the agent's base:

*MEM case:* for each transition, the environment translates between the collection of $(\mathbf{m}_t, \mathbf{s}_t)$ and $(\mathbf{m}_{t+1}, \mathbf{s}_{t+1})$. This ensures that the sensorimotor experiences collected by the agent do not fulfill condition I, and a fortiori condition II.

*MM case:* the environment never translates and keeps its initial position during the collection of all sensorimotor transitions. This ensures that the sensorimotor experiences collected by the agent fulfill condition I, but not condition II.

*MME case:* for each transition, the environment translates after the collection of both $(\mathbf{m}_t, \mathbf{s}_t)$ and $(\mathbf{m}_{t+1}, \mathbf{s}_{t+1})$. This ensures that the sensorimotor experiences collected by the agent fulfill conditions I and II.

Before being fed to the network, the whole collected dataset is normalized such that each motor and sensory component spans $[-1, 1]$ over the whole dataset (performed independently for each simulation). For each training epoch, 100 quadruplets $(\mathbf{m}_t, \mathbf{s}_t, \mathbf{m}_{t+1}, \mathbf{s}_{t+1})$ are randomly drawn in the dataset to form a mini-batch.

## B.4 Remarks

The overall neural network architecture and training procedure have been kept simple. No particular heuristics have been added to improve convergence, generalization, or any other property of the network such as sparsity. Similarly, the architecture's meta-parameters have not been optimized beyond simply checking that the network was expressive enough to approximate the expected mappings. The same size of $\text{Net}_{\text{enc}}$ and $\text{Net}_{\text{pred}}$ have for instance been used in both simulations, even if the sensorimotor mapping is significantly more complex in the Arm in a room simulation than in the Discrete world simulation. Moreover, the network's generalization capacity has not been evaluated. This is because the prime goal of this work is not to optimize a neural network to efficiently solve a task, but rather to study if spatial invariants are captured as a byproduct of sensorimotor prediction, without the need for additional priors.

The code to generate the sensorimotor data, train the neural network, and analyze the motor representation is available at `https://github.com/alaflaquiere/learn-spatial-structure`.

## C  Simulations

Two different agent-environment systems are simulated to generate sensorimotor experiences:

*Discrete world*: This corresponds to an artificial setup designed to optimally evaluate the impact of the experience of sensorimotor invariants on the motor representation built by the network (no

Figure 8: Examples of sensory states received by the agent in the Arm in a room simulation. The spatial and RGB structure of each image is discarded before being fed to the network by flattening its $16 \times 16 \times 3$ values into a simple vector of length 768. (Best seen in color)

sensory ambiguity, continuous sensorimotor mapping, no border effect).

The environment consists in a grid world of size $10 \times 10$. Each square of the grid is associated with a sensory state $\mathbf{s}$ of dimension $N_s = 4$ that a sensor can capture. This sensory state is set to vary smoothly with the position $(r, c)$ of the square in the grid. To ensure such a smoothness, each component $s_i$ of the sensory state $\mathbf{s} = [s_1, s_2, s_3, s_4]$ is defined as a sum of random periodic functions varying with respect to $r$ or $c$:

$$s_i = \sum_k^3 \frac{1}{\lambda_{1,k}^i} \cos\left(2\pi\left(\lfloor\lambda_{1,k}^i\rceil\frac{r}{10} + \lambda_{2,k}^i\right)\right) + \frac{1}{\lambda_{3,k}^i}\cos\left(2\pi\left(\lfloor\lambda_{1,k}^i\rceil\frac{c}{10} + \lambda_{4,k}^i\right)\right), \qquad (14)$$

where all $\lambda$ parameters are randomly drawn in $[-2, 2]$, and $\lfloor . \rceil$ denotes the rounding operation necessary to ensure that the frequency of the function is a multiple of the size of the grid.

The agent's base (invisible for the sensor) can be placed in any square of the grid. The agent has a sensor that it can move in a working space of size $5 \times 5$ centered on its base. In each square, the sensor receives the corresponding sensory state $\mathbf{s}$. To change its sensor position, the agent generates motor states $\mathbf{m} = [m_1, m_2, m_3]$ of dimension $N_m = 3$. Each motor state $\mathbf{m}$ is associated with an egocentric position $\mathbf{p}$ of the sensor in the working space in the following way:

$$\mathbf{p} = 4 \times [\sqrt[3]{m_1}, \sqrt[3]{m_2}] - 2, \qquad (15)$$

where each $m_i$ lives in $[0, 1]$. This arbitrary forward mapping is purposefully made non-linear and redundant, as $m_3$ does not affect the sensor position. Because of this non-linearity and the discrete nature of the grid world, the agent can only sample its motor space in a non-linear fashion (see Fig. 9). Note that this forward mapping is artificial, and we did not define any actual physical body to instantiate it.

Finally, the environment can translate rigidly with respect to the agent's base, effectively moving the working space in the whole grid. The amplitude of this translation is drawn uniformly in $[-10, 10]$ for both its horizontal and vertical components. The whole grid world is set to act as a torus, which means that the sensor appears on the other side of the grid displayed in Fig. 1 when the working space extends beyond its limits.

*Arm in a room*: This corresponds to a more complex and realistic setup in which an arm explores a 3D room. The environment is similar to the one proposed in [9]. The room is of size $7 \times 7$ units, has walls, and is filled with 16 random simple geometric objects. The textures and colors of the walls/floor and objects are picked randomly at the beginning of the simulation. The objects are distributed along a regular grid, but disturbed with an additional small displacement drawn in $U(-0.3, 0.3)^2$ in order to add some randomness.

The agent is a three-segment arm moving in the horizontal plane at a height of 1.6 units. It is equipped with a RGB camera of resolution $16 \times 16$ at its end, orientated with a downward tilt of 0.62 rad, and generating a sensory state $\mathbf{s}$ of dimension $N_s = 16 \times 16 \times 3 = 768$. Note that due to its orientation, the sensor can see the objects in the room but not the arm segments (see Fig. 8). The three arm segments are each of length 0.5 unit. Their respective relative orientation are controlled in $[-\pi, \pi]$ radians by three independent components of the motor state $\mathbf{m}$ of dimension $N_m = 3$. The effective working space of the agent is thus an horizontal disk of diameter 3 units. During the arm movements, the orientation of the sensor is kept fixed.

Finally, the environment can translate rigidly with respect to the arm's base with a maximal horizontal and vertical range of $[-1.75, 1.75]$, where a translation of $[0, 0]$ corresponds to the room being centered on the agent's base. The cumulative effect of the agent's movements and the environment's movements is such that the sensor never moves outside the walls of the room.

Figure 9: Visualization of the normalized regular motor sampling $\mathbf{m}$ (blue dots), its representation $\mathbf{h}$ in the representational space (red dots), and the corresponding ground truth position $\mathbf{p}$ (blue circles) for the three types of exploration in the Discrete world simulation. The affine projection $\mathbf{h}^{(p)}$ of the representations are also displayed in the space of positions. Lines have been added to visualize the distances between each $\mathbf{h}^{(p)}$ and its ground truth counterpart $\mathbf{p}$. Finally, the predicted sensory states $\tilde{\mathbf{s}}_{t+1}$ (magenta dots) outputted by the network are displayed in the 3 first dimensions of the sensory space, alongside the ground-truth sensory states $\mathbf{s}_{t+1}$ (green circles). (Best seen in color)

# D   Detailed results analysis

The results of Fig. 2 and Fig. 3 are here analyzed in more details.

### D.1  Discrete world

**MEM exploration:**  One can see in Fig. 2 that the loss stays relatively high during the whole training. This is due to the fact that it is impossible to accurately predict the future sensory input $\mathbf{s}_{t+1}$ as the environment is always moving between $t$ and $t + 1$. As a consequence, the network learns to output the average sensory state which minimizes the MSE (see the sensory space in Fig. 9).

The topological dissimilarity $D_{topo}$ also stays at a relatively high value during training, and displays an important standard deviation. For each run, $D_{topo}$ varies greatly during the whole training, although it tends to stabilize after $8 \times 10^4$ epochs, when the learning rate reaches its minimum value. This behavior seems to indicate that the topology of $\mathbf{h}$ differs from the one of $\mathbf{p}$, and that the network tends to build an arbitrary motor representation. This is confirmed in Fig. 9, where both $\mathbf{h}$ and its affine projection $\mathbf{h}^{(p)}$ in the space of positions display an arbitrary topology compared to the one of the ground-truth $\mathbf{p}$.

The metric dissimilarity $D_{metric}$ also stays at a relatively high value during training, and displays an important standard deviation. Once again, $D_{metric}$ varies greatly for each run, but stabilizes a bit after $8 \times 10^4$ epochs. Just like for the topology, this seems to indicate that the metric of $\mathbf{h}$ differs from the one of $\mathbf{p}$, which is confirmed in Fig. 9.

**MM exploration:**  One can see in Fig. 2 that the loss quickly converges to very small values. It is expected, as a static environment ensures that the network can easily learn to map the motor states to their corresponding sensory states. This is confirmed in Fig. 9 where the future sensory states appear to be accurately predicted.

The topological dissimilarity $D_{topo}$ also quickly converges to very small values. This seems to indicate that the topology of $\mathbf{h}$ is similar to the one of $\mathbf{p}$. This is confirmed in Fig. 9, where one can see that all redundant motor states associated with the same sensor position are encoded with the same representation, and that the global topology of the manifold of representations is equivalent to the one of the ground-truth position. The manifold of motor encoding is thus practically of dimension 2, when the motor space is actually of dimension 3.

The metric dissimilarity $D_{metric}$ converges to an average value, between $0$ and its value in the MEM case. Its standard deviation is also significant. This seems to indicate that the metric of $\mathbf{h}$ differs from the one of $\mathbf{p}$, which is confirmed in Fig. 9. This lower value of $D_{metric}$ compared to the MEM case is however due to the fact that capturing the topology of $\mathbf{p}$ in $\mathbf{h}$ necessarily entails that the metric difference between the two is lower than compared to a random projection.

**MME exploration:**  Just like in the MM case, the loss quickly converges to very small values as the consistency of the sensorimotor transitions ensures that the network can predict the future sensory state based on the current sensorimotor pair. This is once again confirmed in Fig. 9 where the future sensory states are accurately predicted by the network.

The topological dissimilarity $D_{topo}$ also quickly converges to very small values. This seems to indicate that the topology of $\mathbf{h}$ is similar to the one of $\mathbf{p}$. This is confirmed in Fig. 9, where one can see that all redundant motor states associated with the same sensor position are encoded with the same representation, and that the global topology of the manifold of representations is equivalent to the one of the ground-truth position.

Finally, the metric dissimilarity $D_{metric}$ converges to a very small value, with a very small standard deviation. This seems to indicate that the metric of $\mathbf{h}$ is similar to the one of $\mathbf{p}$, which is confirmed in Fig. 9. In particular, we can see that $\mathbf{h}^{(p)}$ perfectly aligns with the grid of ground-truth positions. Thus there exists a simple affine transformation between the motor representation built by the network and the external position of the sensor.

### D.2  Arm in a room

The analysis done for the Discrete world also globally applies to the Arm in a room simulation. We thus only focus on the differences below.

**Expressivity of Net$_{\mathbf{pred}}$:**  The sensorimotor mapping to learn is significantly more complex in the Arm in a room simulation, and the sensory space is of significantly higher dimension. As a

Figure 10: Visualization of the normalized regular motor sampling **m** (blue dots), its representation **h** in the representational space (red dots), and the corresponding ground truth position **p** (blue circles) for the three types of exploration in the Arm in a room simulation. The affine projection $\mathbf{h}^{(p)}$ of the representations are also displayed in the space of positions. Lines have been added to visualize the distances between each $\mathbf{h}^{(p)}$ and its ground truth counterpart **p**. Finally, the predicted sensory states $\tilde{\mathbf{s}}_{t+1}$ (magenta dots) outputted by the network are displayed in the 3 first dimensions of the sensory space, alongside the ground-truth sensory states (green circles). (Best seen in color)

consequence the neural network, and in particular the $\text{Net}_{\text{pred}}$ module, is not expressive enough to perfectly predict future sensory states, even in cases when it should be theoretically possible (MM and MME cases). This results in the loss and its standard deviation being of greater amplitude than in the Discrete world simulation. This phenomenon is also illustrated in Fig. 10, as one can see in the sensory space that the predictions outputted by the network do not perfectly match the ground-truth in the MM and MME cases.

Figure 11: Visualization of the normalized regular motor sampling **m** (blue dots), its representation **h** in the representational space (red dots), and the corresponding ground truth position **p** (blue circles) for an Arm in a room simulation and an MME exploration. The affine projections $\mathbf{h}^{(\mathbf{p})}$ are also displayed in the space of positions. In this trial, the manifold of **h** slightly spread along a third dimension in the representational space. (Best seen in color)

Note that this lack of expressivity could be seen as a shortcoming of our evaluation. But on the contrary we consider it as a highlight, as it shows that the motor representation tends to capture topological and metric invariants even when the overall performance of the sensorimotor prediction is limited.

**Limited exploration:** An important difference between the Discrete world and Arm in a room simulations is that the exploration is limited in the latter. Indeed, in the former, the amplitude of environmental translations, coupled with the fact that the grid world acts as a torus, ensures that any motor state can be associated with the sensor being in any square of the grid. As a consequence, all motor states are statistically associated with the same distribution of sensory states over the whole environment. On the contrary, in the Arm in a room simulation, the environment is limited by walls. As a consequence, each motor state only covers a sub-part of the environment when the latter moves. For instance, a motor state corresponding to the arm being extended to the left means that the sensor will never experience the sensory states on the far right of the room. Thus each motor state is associated with a slightly different sensory distribution when the environment moves. It is thus possible for the network to infer an approximation of the topology of **p** which helps to reduce the MSE, even in the MEM case. This can be seen in Fig. 10 as the motor representation tends to capture the topology of **p** in the MEM case. This effect is also visible in the sensory space as one can see that, in this simulation, the network does not simply output the same average sensory state for all motor states. Instead, each motor state is associated with the average sensory state over the slightly different sensory distribution is its associated with.

The limited exploration thus impacts the measures $D_{topo}$ and $D_{metric}$ in the MEM case. They are lower than what could be expected if we simply extrapolated from the results of the previous simulation.

**Ambiguous environments:** The third difference is that the 3D room environments can present some sensory ambiguities: very different sensor positions can be associated with very similar sensory states. As a consequence, the sensory manifold associated with the manifold of positions can be twisted in a non-trivial way in the sensory space. In case of perfect ambiguity between two (or more) positions, the topology of the manifold even changes locally. This perturbs the learning of the motor representation when the environment is static (MM case), and leads to $D_{topo}$ measures greater than what could be expected if we simply extrapolated from the results of the previous simulation. Note however that this sensory ambiguity is not a problem in the MEM case, as the movements of the environment ensure that the different sensor positions which are ambiguous for a given environment position are associated with different sensory distributions over the whole exploration.

**Non-flat representations:** The final difference is the one that leads to the standard deviation of $D_{topo}$ being more important than expected in the MME case. As can be observed in Fig 11, it sometimes happens in the MME case that the manifold of **h** appears slightly spread in the 3D representational space instead of approximating a 2D flat manifold. Yet, the affine projection of **h** in

Figure 12: Evolution of the loss and the dissimilarity measures $D_{topo}$ and $D_{metric}$ during training for both setups, for the three types of exploration, and with $N_h = 25$ instead of 3. The displayed means and standard deviations are computed over 50 independent runs. (Best seen in color)

the space of $\mathbf{p}$ properly aligns with the ground-truth positions. This phenomenon is due to the way a neural network processes data. Indeed the representation $\mathbf{h}$ is fed to a fully connected layer where each neuron performs a linear projection of $\mathbf{h}$ before passing it through its activation function. As a consequence, the network has two equivalent options to respect the sensory invariants induced by the MME exploration: i) flatten (to 2D) the manifold in the 3D representational space, or ii) tune the weights to the next fully connected layer such that all neurons perform projections which take into account only two dimensions in the representational space. In both cases, the input received by the predictive module $\text{Net}_{\text{enc}}$ is equivalent. This also explains why even a non-flat manifold of $\mathbf{h}$ still matches the ground-truth position, as the linear regression correspond to a projection of the same nature as the one implemented by the connections to the next layer. The effect of such a phenomenon can also be seen in Fig. 2, as it explains why $D_{topo}$ shows a significant standard deviation in the MME case.

# E    Additional experiments

## E.1    Representational space of higher dimension

The main results were obtained with a representational space $\mathbb{R}^{N_h}$ of dimension $N_h = 3$ to facilitate visualization. The same exact experiments were also run with a representational space of dimension $N_h = 25$. The results, presented in Fig. 12, are qualitatively equivalent to the ones described previously[2]. This seems to indicate that the dimension of the representational space has no influence on the way space-induced invariants are captured in $\mathbf{h}$.

## E.2    Forward mappings of higher complexity

The robustness of the results with respect to the complexity of the agent's body has been evaluated by designing more complex forward mappings. In the Discrete world setup, we designed a new agent with $N_m = 6$ motors and the following forward mapping:

$$\mathbf{p} = \begin{bmatrix} x \\ y \end{bmatrix} = 4 \times (P \cdot A \cdot f(\mathbf{m})) - 2, \ \text{with:} \ f\left(\begin{bmatrix} m_1 \\ m_2 \\ m_3 \\ m_4 \\ m_5 \\ m_6 \end{bmatrix}\right) = \begin{bmatrix} m_1^2 \\ \sqrt{m_2} \\ \sqrt[3]{m_3} \\ 0.1 \times \left(\frac{1.1}{0.1}\right)^{m_4} - 0.1 \\ \log(m_5 \times (e^1 - 1) + 1) \\ m_6 \end{bmatrix}, \quad (16)$$

where $A$ is a $6 \times 6$ mixing matrix with random elements uniformly drawn in $[-2, 2]$, and $P$ is a diagonal projection matrix whose first two elements are equal to $1$ and the others to $0$. The elements of $\mathbf{m}$ are sampled from $[0, 1]$ and mapped to $[0, 1]$ by the non-linear transformation $f$. The random

matrix $A$ is also carefully designed such that the mixed components of $A \cdot f(\mathbf{m})$ still belong to $[0, 1]$. Finally, a simple linear transformation is applied after the projection $P$ such that the sensor coordinates $x$ and $y$ lie in $[-2, 2]$. Intuitively, the overall transformation consists in passing the 6 motor components through some non-linearities, mixing them via a random matrix $A$, and finally projecting the result in 2D via the matrix $P$.

In the Arm in a room setup, we designed a four-segment arm agent with $N_m = 6$ motor:, four hinge joints, and two translational joints on the central two segments. The arm still moves in the horizontal plane, but the (maximum) length of its segments has been reduced to 0.375 so that the working space's radius is unchanged. The hinge joints are identical to the previous agent, and translational joints are controlled in [-0.375, 0.375].

Due to the increased complexity of the forward mapping to estimate, we increased the size of the $\mathrm{Net}_{\mathrm{enc}}$ module to $(500, 400, 300, 200)$ (in both setups), and increased the number of collected exploratory transitions to 300000 instead of 150000 (in the Arm in a room setup).

The results of the experiments with these more complex forward mappings are presented in Figs. 13, 14, and 15. They are qualitatively equivalent to the ones observed with the original forward models. This seems to indicate that the complexity of the forward model (body of the agent) has no influence on the way space-induced invariants are captured in $\mathbf{h}$.

We can however note some quantitative difference in Fig. 13. Indeed, due to the increased dimension of the motor space, the regular sampling performed during the network evaluation requires significantly more motor samples (15625 instead of 125 and 216 in the original Discrete world and Arm in a room setups respectively). The loss is thus computed over more samples. The higher number of degrees of freedom also means a higher degree of redundancy. As a result, more motor states are associated with the same egocentric position of the sensor. This has an indirect effect on the values observed for $D_{topo}$ and $D_{metric}$, as the arbitrary encoding induced by the MEM exploration results in significantly higher dissimilarity measures. A log scale has then been used in Fig. 13 in order to better visualize the results.

Figure 13: Evolution of the loss and the dissimilarity measures $D_{topo}$ and $D_{metric}$ during training for both setups, for the three types of exploration, and with the more complex agents (6 degrees of freedom). A log-scale is used on the y-axis due to the large values induced by the MEM exploration. The displayed means and standard deviations are computed over 50 independent runs. (Best seen in color)

### E.3 Intrinsic stochasticity of the training

Due to the stochasticity of the training procedure (network initialization, mini-batches selection), the learning curves display some intrinsic variability, even when trained on a fixed dataset. We estimated this variability and display in Fig. 16 the average and standard deviation associated with 50 independent runs trained on the same dataset, for each simulation.

Compared to Fig. 2, one can see that the resulting standard deviations are very similar. This seems to indicate that most of the variability observed in the results is due to the intrinsic variability of the training procedure, rather than to the datasets on which the networks are trained.

Figure 14: Visualization of the normalized regular motor sampling $\mathbf{m}$ (blue dots) in the 3 first dimensions of the 6-D motor space, its representation $\mathbf{h}$ in the representational space (red dots), and the corresponding ground truth position $\mathbf{p}$ (blue circles) for the three types of exploration in the Discrete world simulation with the more complex agent. The affine projection $\mathbf{h}^{(p)}$ of the representations are also displayed in the space of positions. Lines have been added to visualize the distances between each $\mathbf{h}^{(p)}$ and its ground truth counterpart $\mathbf{p}$. Finally, the predicted sensory states $\tilde{\mathbf{s}}_{t+1}$ (magenta dots) outputted by the network are displayed in the 3 first dimensions of the sensory space, alongside the ground-truth sensory states (green circles). (Best seen in color)

Figure 15: Visualization of the normalized regular motor sampling **m** (blue dots) in the 3 first dimensions of the 6-D motor space, its representation **h** in the representational space (red dots), and the corresponding ground truth position **p** (blue circles) for the three types of exploration in the Arm in a room simulation with the more complex agent. The affine projection $\mathbf{h}^{(p)}$ of the representations are also displayed in the space of positions. Lines have been added to visualize the distances between each $\mathbf{h}^{(p)}$ and its ground truth counterpart **p**. Finally, the predicted sensory states $\tilde{\mathbf{s}}_{t+1}$ (magenta dots) outputted by the network are displayed in the 3 first dimensions of the sensory space, alongside the ground-truth sensory states (green circles). (Best seen in color)

Figure 16: Evolution of the loss and the dissimilarity measures $D_{topo}$ and $D_{metric}$ during training for both setups, for the three types of exploration, and with $N_h = 3$. The displayed means and standard deviations are computed over 50 independent runs trained on a single dataset for each simulation. (Best seen in color)

## Footnotes

[1]Note that this assumption would not hold for $g$, as suggested by Eq. (3).

[2]The higher dimension also implies an even bigger effect of the potential twist of the sensory manifold and of the spread of the representation in more than 2 dimensions (see Sec D.2).