[Reviews · NeurIPS 2019]

Reviewer 1



For me is not clear what is the advantage of this model compared to the many previous work using self organizing maps, predictive coding or slow feature analysis. Nowadays there is a lot of work also in unsupervised learning depth from images.

Reviewer 2



1. I wonder how does the behavior change if the morphology of the agent (motor configurations) is made more complex. If I understand correctly the agent only has 3 degrees of freedom (Figure 3 (left))? What would happen if the agent were much more complex, say with very many degrees of freedom. Will we still see the same behavior as in the current experiments in the paper?

Reviewer 3



Originality This paper is fairly original as it formalises the problem of sensorimotor prediction and evaluation. Self-localization is common topic that is often addressed in RL and neuro-science, however it was nice to see a proper analysis of the problem together with the invariant and how to measure them. Quality Overall I think this paper is of high quality and is nicely self-contained. The authors motivate the problem well, give a formal declaration of the problem setup, define the invariant properties that we are looking for in agents as proof of sensorimotor awareness, then present the loss that is supposed to make this behaviour emerge, present ways of measuring the sensorimotor awareness via invariants, and finally run experiment and show positive results via 3 different environment setups. Clarify Even thought he paper is well written I think it could be clearer. I feel like the authors complicate the problem by introducing a lot of different notations and a very generic definition of the problem - maybe a running example throughout the manuscript would have helped to understand the concepts faster. The paper boils down to this: the hypothesis is that performing next step prediction of some carefully chosen values makes self localization emerge under the right environment properties. I think this message needs to be stated in a simple way as it is easy for the reader to get lost in the details. A diagram for the two metrics would have been useful too. Significance Overall I feel that it is a positive, sensible result, however I am not sure how significant this result is. It seems natural and obvious that such properties would emerge in the given setup, but it is also nice to have some work that formally present and show that it is actually the case. I do not know the literature well enough to judge whether this has been shown before or not, however I would have liked to see in the related work section a justification of why this paper is unique and what it brings compared to other works.

Reviewer 4



Originality: Moderate. This is an area of research that has received a significant amount of attention, as the authors indicate. But the specific investigation in this work regarding the sufficient conditions for certain structural properties to emerge is useful and novel. Quality: Good. The approach appears to be technically sound. The claims are supported by thorough variation of the environmental properties involved. Clarity: Good. The paper is clearly written and very well organized. One moderate weakness in the exposition is the treatment of Condition II regarding metric structure. If I understand correctly, the environmental conditions that are sufficient for this property to emerge amount to "dataset completeness" in the form of the agent observing transitions under a wide variety of environmental configurations. The structure learned under violation of this condition can be seen as "overfitting" to the particular environmental configuration (as it is static) and therefore is only topologically similar to the true state. This is somewhat subtle and could use further exposition. Significance: Moderate. This is important work, and is likely to provide a basis for future researchers in this area to analyze and develop environments to investigate and encourage the emergence of spatial representations. The architecture proposed in this work is not a significantly useful contribution and the results are not immediately illuminating about any particular application domain, but the lessons learned will be valuable to future researchers.

[Author Response · NeurIPS 2019]

We thank the Reviewers for their constructive comments that helped improve the paper. We address their comments and questions below, and modified the manuscript accordingly.

Reviewer 3 raised the question of the generalization to more complex agent morphologies, which was not addressed in the submission. Following their suggestion, we ran additional experiments with higher dimensional motor spaces (6D) and more complex forward models: i) involving non-linear functions and random linear combinations of the motor command (discrete world), and ii) a 4-segment arm with 4 hinge and 2 translational joints (arm in a room). The new results are qualitatively equivalent to the ones presented in the submission, which indicate that the capture of spatial invariants is robust to the complexity of the agent. Due to compute limitations, only preliminary results are currently available and have been added in the Appendix. They will be updated as soon as all simulations have finished running. (Remark: for the new arm, we collected more samples and increased the size of $\text{Net}_{\text{enc}}$ to be able to approximate the more complex forward mapping.)

Following the suggestions of Reviewers 3 and 4, we extended the description of the problem setup to better ground the notations and concepts. We also added diagrams in the Appendix to illustrate i) the connection between our generic formalism and the simulations, and ii) the spatial invariants, that were previously only described mathematically.

Following Reviewer 4's recommendation, we added a simple and concise description of the paper's aim at the end of the Related work section: *This work is in line with the theoretical developments of [...], which address the fundamental problem of space perception in the framework of the SMCT, but frame them in an unsupervised machine learning framework. We show that the structure of space can get naturally captured as a by-product of sensorimotor prediction.*

Regarding the additional comments of Reviewers 1 and 4, it might be important to point out that our perspective significantly differs from the bulk of literature on spatial representation. Our aim is not to solve a task or to compete over performance measures but, more fundamentally, to study how the concept of *space* can emerge in a naive agent. Thus a direct performance comparison with other methods/tools (self-organizing maps, slow feature analysis...) is of limited interest. (Additionally, depth estimation is a problem of a different nature than the one addressed in this work). We can however say that most approaches to this problem build allocentric representations, based primarily on sensory information (with occasional auxiliary interactions with motor information), and using a priori defined constraints to shape the representation. These representations are build either to solve a task, in which case their structure is of secondary importance, or enforced to be spatial-like via prior constraints. As a consequence, their structure at most captures the topology of a sensory manifold (equivalent to the topology of space for a single environmental state only) or one imposed from the motor space via additional prior constraints. In contrast, we study the conditions for the emergence of an egocentric spatial representation, based on the fundamental study of how an external space induces invariants in an agent's sensorimotor experience (originating from the concept of *compensability* introduced by H.Poincaré). This means describing how the very structure of space shapes sensorimotor experiences, and how it impacts sensorimotor prediction, in the absence of any specific task or extraneous constraint. This way, the approach can be seen as in agreement with the SMCT and Predictive Coding. Moreover, the nature of the spatial invariants we identified means that our representation is grounded in the motor space, abstracted from the specific content of the environment, and captures the metric structure of space; a result that no other non-supervised method achieves. Due to its originality, only few previous works align directly with this line of research. They are all mentioned in the introduction and related work of the original submission (refs 10, 23-27, 35, 36, 38). This work is the first to frame their theoretical considerations about spatial invariants into a self-supervised machine learning framework. We updated the end of the introduction to better emphasize the nature of this contribution, and significantly extended the related work section to better explain how our approach differs from the cited works.

We include below a (tiny) overview of the five diagrams and figures that were added to the paper:

Figure 1: From left to right: Partial view of the problem setup diagram; Partial view of the invariants diagrams; MME results for the more complex discrete agent; MME results for the more complex arm in a room.

[Meta-Review · NeurIPS 2019]

We had quite an extensive discussion about this paper after the author response. The reviewers appreciated the clarifications and especially the additional experiment. What makes the paper stand out is proposing two generic conditions that enforce the emergence of spatial structures and experimentally validating them. The discussion circled around Equation (1), how well that would hold for realistic (noisy) sensing and what the implication on the emergence of the spatial structures would be. To our understanding the later parts of the paper don't rely on this equation but only on the intuition. Nevertheless, it would be nice to include a discussion on the impact of realistic sensing. A few related papers that were mentioned by R1: [0] D. Pierce and B. Kuipers, "Map learning with uninterpreted sensors and effectors.," Artificial Intelligence , vol. 92, pp. 169–229, 1997. [1] J. Modayil, "Bootstrap Learning a Perceptually Grounded Object Ontology." 2004. [2] J. M. Benjamin Kuipers Patrick Beeson, "Bootstrap learning of foundational representations.," Connection Science , vol. 18, no. 2, pp. 145–158, Jun. 2006. [3] J. Modayil and B. Kuipers, "Autonomous Development of a Grounded Object Ontology by a Learning Robot.," in Proceedings of the Twenty-Second National Conference on Artificial Intelligence (AAAI-07) , 2007. [4] J. Modayil, "Discovering sensor space: Constructing spatial embeddings that explain sensor correlations," in Development and Learning (ICDL), 2010 IEEE 9th International Conference on , 2010, pp. 120–125. [5] A. M. Dearden and Y. Demiris, "Learning Forward Models for Robots.," in IJCAI-05, Proceedings of the Nineteenth International Joint Conference on Artificial Intelligence, Edinburgh, Scotland, UK, July 30-August 5, 2005 , 2005, pp. 1440–1445. [6] Y. Demiris and A. Dearden, "From motor babbling to hierarchical learning by imitation: a robot developmental pathway," in Proceedings Fifth International Workshop on Epigenetic Robotics: Modeling Cognitive Development in Robotic Systems , 2005, pp. 31–37. [7] D. Ognibene, F. Mannella, G. Pezzulo, G. Baldassarre. "Integrating Reinforcement-Learning, Accumulator Models, and Motor-Primitives to Study Action Selection and Reaching in Monkeys," Proceedings of the 7th International Conference on Cognitive Modelling, 2006 [8] D. Ognibene, A. Rega, G. Baldassarre, "A model of reaching that integrates reinforcement learning and population encoding of postures," From Animals to Animats 9: Proceedings of the Ninth International Conference, 2006